# Fisher and Shannon Functionals for Hyperbolic Diffusion

**DOI:** 10.3390/e25121627

**Published:** 2023-12-06

**Authors:** Manuel O. Cáceres, Marco Nizama, Flavia Pennini

**Affiliations:** 1Comision Nacional de Energia Atomica, Centro Atomico Bariloche and Instituto Balseiro, Universidad Nacional de Cuyo, Av. E. Bustillo 9500, Bariloche CP 8400, Argentina; 2CONICET, Centro Atomico Bariloche, Av. E. Bustillo 9500, Bariloche CP 8400, Argentina; 3Departamento de Fisica, Facultad de Ingenieria and CONICET, Universidad Nacional del Comahue, Neuquen CP 8300, Argentina; marconizama@gmail.com; 4Departamento de Física, Facultad de Ingeniería, Universidad Nacional de Mar del Plata (UNMDP), CONICET, Mar del Plata CP 7600, Argentina; flavia.pennini@gmail.com; 5Departamento de Física, Universidad Católica del Norte, Av. Angamos 0610, Antofagasta 1270709, Chile

**Keywords:** hyperbolic diffusion, telegrapher’s equation, Shannon entropy, Fisher information, Cramer-Rao bound

## Abstract

The complexity measure for the distribution in space-time of a finite-velocity diffusion process is calculated. Numerical results are presented for the calculation of Fisher’s information, Shannon’s entropy, and the Cramér–Rao inequality, all of which are associated with a positively normalized solution to the telegrapher’s equation. In the framework of hyperbolic diffusion, the *non-local* Fisher’s information with the *x*-parameter is related to the *local* Fisher’s information with the *t*-parameter. A perturbation theory is presented to calculate Shannon’s entropy of the telegrapher’s equation at long times, as well as a toy model to describe the system as an attenuated wave in the ballistic regime (short times).

## 1. Introduction

Advances in understanding wave propagation in a conducting medium were achieved through the analysis of the telegrapher’s equation (TE), which originally appeared in the study of electromagnetic fields in waveguides [1,2,3]. This hyperbolic diffusion equation has been used in different areas of research, including the hyperbolic heat equation [4], generalized Cattaneo–Fick equations [5,6], neuroscience [7,8], biomedical optics [9], electromagnetic analysis in multilayered conductor planes [10], penetration of waves in complex conducting media [11], asymptotic diffusion from Boltzmann anisotropic scattering [12,13,14], TE in 2D and 3D for engineers problems [15], describing cosmic microwave background radiation with spherically hyperbolic diffusion [16,17], finite-velocity diffusion in heterogeneous media [18,19,20,21], as well as in the damping and propagation of surface gravity waves on a random bottom [22].

The study of the propagation of thermal waves is also a fundamental theoretical and experimental subject [23,24], where the temperature profile $\psi \left(x,t\right)>0$ can be described by the TE:(1)$$\left[\partial_{t}^{2}+\frac{1}{\tau }\partial_{t}-v^{2}\partial_{x}^{2}\right]\psi \left(x,t\right)=0,$$
thus, the wave packet propagates at the velocity (*v*) and is attenuated at a rate of $\tau^{-1}$. The same TE serves as the starting point for studying electromagnetic field transport in waveguides, where Ohm’s law plays a fundamental role in describing the conducting media and characterizing the dissipative parameter $\tau^{-1}$ [1,2]. In this case, an electromagnetic dissipative wave is the solution to (Equation 1), but it is not necessary to impose positivity and normalization on it. It is interesting to note that Equation (Equation 1) has two extreme cases:The wave limit

Taking τ→∞, we recover the wave equation: $\left[\partial_{t}^{2}-v^{2}\partial_{x}^{2}\right]\psi_{\mathrm{WE}}\left(x,t\right)=0$. Then, its solution can then be considered a wave packet that moves either to the right or left, represented as $\psi_{\mathrm{WE}}\left(x\pm vt\right)$, without changing its form throughout the whole domain.
The diffusion limit

Taking τ→0 and v→∞, such that $\tau v^{2}\to D$, we recover the diffusion equation: $\left[\partial_{t}-D\partial_{x}^{2}\right]\psi_{W}\left(x,t\right)=0$, where its solution is given by the Wiener process: $\psi_{W}\left(x,t\right)=\exp\left(-x^{2}/4Dt\right)/\sqrt{\pi 4Dt}$.

In the following sections, we will present a Shannon entropic analysis [25] for the TE. Fisher’s information [26] is naturally linked to variations of entropy; therefore, it can be related to the control of disorder in a transport process. Fisher’s measure of indeterminacy has several natural and important applications in the design of codes and protocols, biophysics transport, machine learning, etc. In fact, this measure tells us how much information one (input) parameter carries about another (output) value, and so Fisher’s information is widely used in optimal experimental design in many areas of research [27,28]. In the next sections, the analysis of the Shannon and Fisher functionals will be presented for the finite-velocity diffusion process (hyperbolic diffusion).

The paper is organized as follows: In Section 2, we present hyperbolic diffusion as a novel integro-differential equation for a non-Markov diffusion process. In Section 3, we present, for the first time, to our knowledge, the Fisher and Shannon functionals for the hyperbolic diffusion problem; in the Markovian limit, Fisher and Shannon functionals for Wiener’s diffusion are recovered. In Section 4, we calculate Shannon’s entropy, Fisher’s information, the complexity measure, and the Cramér–Rao bound. All of these novel results are obtained by solving numerically the TE (Equation 1). In Section 5, we present conclusions on the present approach, as well as its future extensions and applications. In addition, Appendix A, Appendix B and Appendix C are used to present the algebra necessary to prove the diffusion convergence, introduce a novel toy model for short times, and develop a perturbation theory for the calculation of Shannon’s entropy at long times.

## 2. Hyperbolic Diffusion

In the following, we are concerned with the hyperbolic diffusion problem. Therefore, in the present work, we are interested in solutions to (Equation 1) under the following conditions:(2)$$\psi \left(x,t\right)\geq 0,\;\int_{-\infty }^{\infty }dx\;\psi \left(x,t\right)=1,$$
and we look for a solution that fulfills the particular initial conditions:(3)$${\left.\psi \left(x,t\right)\right|}_{t=0}=\delta \left(x\right),\;{\left.\partial_{t}\psi \left(x,t\right)\right|}_{t=0}=0,$$
where $\delta \left(x\right)$ is the Dirac delta function. An analytic solution can straightforwardly be obtained by working with Equation (Equation 1) in the Fourier–Laplace representation; see Appendix A. Other initial conditions, such as a “bullet packet”, can also be worked out, but will not be considered in the present work.

It is interesting to note that the TE can also be written as an integral operator:(4)$$\partial_{t}\psi \left(x,t\right)=v^{2}\int_{0}^{t}e^{-\left(t-t^{\prime }\right)/\tau }\;\partial_{x}^{2}\psi \left(x,t^{\prime }\right)\;dt^{\prime },\;{\left.\partial_{t}\psi \left(x,t\right)\right|}_{t=0}=0.$$
We note that, from this equation, we cannot use a *bullet initial condition*, because in this case, ${\left.\psi \left(x,t\right)\right|}_{t=0}={\left.\delta \left(x-vt\right)\right|}_{t=0}$ and ${\left.\partial_{t}\psi \left(x,t\right)\right|}_{t=0}\neq 0.$

Equation (Equation 4) demonstrates that the solution to the hyperbolic diffusion can be thought of as a non-Markov diffusion process (see chap. 7 in [29]).

## 3. Fisher and Shannon Functionals

The Shannon entropy can be defined as follows:(5)$$\begin{array}{ccc} S\left(t\right) & = & -\int_{-\infty }^{\infty }dx\;\psi \left(x,t\right)\ln\left[\psi \left(x,t\right)\Delta \right]\\ & = & -\int_{-\infty }^{\infty }dx\;\psi \left(x,t\right)\left[\ln\Delta +\ln\psi \left(x,t\right)\right]\\ & = & \ln\frac{1}{\Delta }-\int_{-\infty }^{\infty }dx\;\psi \left(x,t\right)\ln\psi \left(x,t\right), \end{array}$$
in this formula, we use normalization: $\int_{-\infty }^{\infty }dx\;\psi \left(x,t\right)=1$, and explicitly write the small space parameter Δ, which is necessary for the correct definition of Shannon’s entropy for any distribution fulfilling $\left[dx\;\psi \left(x,t\right)\right]$ = dimensionless. Thus, $\left(S\left(t\right)-\ln\frac{1}{\Delta }\right)\equiv \Delta S\left(t\right)$ can be interpreted as a *relative* entropy to a sharp initial condition at the origin.

### 3.1. Wiener Diffusion Case

Here, we show a relation between the Shannon and Fisher information functionals for the diffusion process. Applying the time derivative to the definition of Shannon’s entropy of the Wiener process, denoted as $S_{W}\left(t\right)$, we obtain:(6)$$\begin{array}{ccc} \frac{dS_{W}\left(t\right)}{dt} & = & -\frac{d}{dt}\int_{-\infty }^{+\infty }dx\;\psi_{W}\left(x,t\right)\;\ln\psi_{W}\left(x,t\right).\\ & = & -\int_{-\infty }^{+\infty }dx\;\ln\psi_{W}\left(x,t\right)\;\partial_{t}\psi_{W}\left(x,t\right), \end{array}$$
where we use the normalization of $\psi_{W}\left(x,t\right)$. For the Wiener process, the distribution fulfills the diffusion equation:$$\partial_{t}\psi_{W}\left(x,t\right)=D\partial_{x}^{2}\psi_{W}\left(x,t\right),$$
then replacing this operator in (Equation 6), we obtain the following expression:$$\frac{dS_{W}\left(t\right)}{dt}=-D\int_{-\infty }^{+\infty }dx\;\ln\psi_{W}\left(x,t\right)\;\partial_{x}^{2}\psi_{W}\left(x,t\right).$$
Integrating by parts on the right-hand side (RHS) and using ${\left[\ln\psi_{W}\left(x,t\right)\;\partial_{x}\psi_{W}\left(x,t\right)\right]}_{x\to \pm \infty }$→0, we obtain the following:(7)$$\begin{array}{ccc} \frac{dS_{W}\left(t\right)}{dt} & = & D\int_{-\infty }^{+\infty }dx\;\partial_{x}\psi_{W}\left(x,t\right)\;\partial_{x}\psi_{W}\left(x,t\right)/\psi_{W}\left(x,t\right)\\ & = & D\int_{-\infty }^{+\infty }dx\;\frac{{\left(\partial_{x}\psi_{W}\left(x,t\right)\right)}^{2}}{\psi_{W}\left(x,t\right)}. \end{array}$$
The RHS expression in (Equation 7) is proportional to Fisher’s information:(8)$$I_{W}\left(t\right)=\langle {\left(\frac{\partial }{\partial_{x}}\ln\psi_{W}\left(x,t\right)\right)}^{2}\rangle ,$$
in this case, Fisher’s information concerns the *x*-parameter, and the symbol $\langle \cdots \rangle$ represents the mean value over the PDF $\psi \left(x,t\right)$. Then, we obtain the following:(9)$$\frac{dS_{W}\left(t\right)}{dt}=D\;I_{W}\left(t\right).$$

Using the Shannon entropy of the Wiener process, $S_{W}\left(t\right)=\ln\sqrt{4\pi eDt}$, in (Equation 9), we arrive at the following:(10)$$I_{W}\left(t\right)=\frac{1}{2Dt},$$
thus, Fisher’s information for the Wiener process is as follows: $I_{W}\left(t\right)={\langle x{\left(t\right)}^{2}\rangle }^{-1}∼t^{-1}$. Then, we write a closed expression, connecting Shannon’s entropy and Fisher’s information for the Wiener process:$$S_{W}\left(t\right)+\frac{1}{2}\ln\left(I_{W}\left(t\right)\right)=\frac{1}{2}\ln\left(2\pi e\right).$$
After a little algebra, we can write the following:(11)$$C_{W}\left(t\right)=\frac{e^{2S_{W}\left(t\right)}I_{W}\left(t\right)}{\ln(2\pi e)}=1,$$
where $C_{W}\left(t\right)$ is the complexity measure. This complexity combines Shannon’s entropy and Fisher’s information for Brownian motion in a simple manner. In general, the complexity measure is defined as follows [27,28]:(12)$$C=\frac{e^{2S}I}{\ln(2\pi e)}\geq 1.$$

In addition, it is simple to see from (Equation 10) that the Cramér–Rao bound [30,31] is fulfilled for the Wiener process. Indeed, using $I_{W}\left(t\right)=1/\langle x{\left(t\right)}^{2}\rangle$ and taking into account the dispersion $\langle x{\left(t\right)}^{2}\rangle =\Delta x{\left(t\right)}^{2}$, we can write the following bound:(13)$$I_{W}\left(t\right)\Delta x{\left(t\right)}^{2}=1,$$
and so we propose a generic Cramér–Rao lower bound to be used in the TE:(14)$$I\left(t\right)\Delta x{\left(t\right)}^{2}\geq 1.$$
The Cramér–Rao inequality is a fundamental tool in information analysis and is widely used in optimal experimental design. Due to the reciprocity of estimator variance and Fisher’s information, minimizing variance corresponds to maximizing information.

In Appendix B, we present the general definition of the θ—parameter Fisher’s information functional.

### 3.2. Hyperbolic Diffusion Case

Here, we present a relationship between the Shannon and Fisher functionals for the solution to the TE. We will show that there is a *non-local* connection between Shannon’s entropy and Fisher’s information. This can be seen by considering that the TE also comes from a non-Markov operator. Using (Equation 4) in the formula for the time derivative of $S_{\mathrm{TE}}\left(t\right)$, we obtain the following:(15)$$\begin{array}{ccc} \frac{dS_{\mathrm{TE}}\left(t\right)}{dt} & = & -\int_{-\infty }^{+\infty }dx\;\ln\psi_{\mathrm{TE}}\left(x,t\right)\;\partial_{t}\psi_{\mathrm{TE}}\left(x,t\right)\\ & = & -\int_{-\infty }^{+\infty }dx\;\ln\psi_{\mathrm{TE}}\left(x,t\right)\;v^{2}\int_{0}^{t}dt^{\prime }\;e^{-\left(t-t^{\prime }\right)/\tau }\;\partial_{x}^{2}\psi_{\mathrm{TE}}\left(x,t^{\prime }\right)\\ & = & v^{2}\int_{0}^{t}dt^{\prime }\;e^{-\left(t-t^{\prime }\right)/\tau }\int_{-\infty }^{+\infty }dx\;\left(\frac{\partial_{x}\psi_{\mathrm{TE}}\left(x,t\right)}{\psi_{\mathrm{TE}}\left(x,t\right)}\right)\partial_{x}\psi_{\mathrm{TE}}\left(x,t^{\prime }\right)\\ & = & v^{2}\int_{0}^{t}dt^{\prime }\;e^{-\left(t-t^{\prime }\right)/\tau }\;I_{\mathrm{TE}}\left(t,t^{\prime }\right), \end{array}$$
the last line is obtained by part integration, and it shows that $dS_{\mathrm{TE}}\left(t\right)/dt$ is connected to the *non-local* Fisher’s information:(16)$$I_{\mathrm{TE}}\left(t,t^{\prime }\right)\equiv \int_{-\infty }^{+\infty }dx\;\frac{\partial_{x}\psi_{\mathrm{TE}}\left(x,t\right)}{\psi_{\mathrm{TE}}\left(x,t\right)}\partial_{x}\psi_{\mathrm{TE}}\left(x,t^{\prime }\right).$$
A connection between the non-local *x*-*parameter* Fisher’s information and the *t*-*parameter* Fisher’s information is demonstrated in Appendix B, as shown in (Equation 40).

As can be seen from (Equation 15), only in the limit τ→0 do we recover a local relation (with v→∞, such that $\tau v^{2}\to D$), then
(17)$$\lim_{\tau \to 0}\tau v^{2}\int_{0}^{t}dt^{\prime }\;\frac{e^{-\left(t-t^{\prime }\right)/\tau }}{\tau }\;I_{\mathrm{TE}}\left(t,t^{\prime }\right)\to DI_{W}\left(t\right).$$
obtaining (Equation 9).

The case τ→∞ must be taken with care because a wave solution $\psi_{\mathrm{WE}}\left(x-vt\right)$ intrinsically represents a bullet-like initial condition. Therefore, the surface terms that come from integration by parts cannot be taken as zero. In addition, as time goes on, for a wave packet that moves without deformation, there must not be an increase in disorder or loss of information. To make this analysis, we can take, asymptotically, the limit t/τ≪1, so from (Equation 15), we obtain the first-order approximation:(18)$$\lim_{t/\tau \ll 1}\frac{dS_{\mathrm{TE}}\left(t\right)}{dt}\to \tau v^{2}\int_{0}^{t/\tau }dz\;e^{z}\;I_{\mathrm{TE}}\left(t,\tau z\right),$$
therefore, in the limit t/τ≪1, we can approximate $S_{\mathrm{TE}}\left(t\right)∼$ as constant in the ballistic regime. See Appendix A.

### 3.3. Estimation Theory

In many experimental situations, a quantity of interest cannot be measured directly; it can only be measured from a sample of data. This situation is covered by the inference theory, sometimes referred to as the estimation approach [32,33]. Regarding a sample of data of independent and identically distributed (i.i.d.) random variables, the Fisher measure provides an invaluable tool for analyzing this problem, particularly the Cramér–Rao lower bound. For stochastic processes, the situation is much more involved. In particular, in the diffusion case, where the increments of the Wiener process are i.i.d. random variables, the analysis is quite accessible, while in hyperbolic diffusion, where there are strong correlations, it is widely unexplored.

To estimate the finite velocity of diffusion “v” in the most precise way possible by sampling from the probability distribution of the hyperbolic diffusion is a very interesting issue. Likelihood estimation in the TE is very important in many experimental situations, and we believe that with the help of these ideas, this analysis will be promoted. Nevertheless, we note that this subject is outside the scope of the present work and will be considered in the future.

## 4. Numerical Results for the Telegrapher’s Equation

Here, we are going to show the numerical results from the direct integration of the TE (Equation 1), and for different values of the parameters $\left\{\tau ,v\right\}$. First, we calculate Shannon’s entropy $S_{\mathrm{TE}}\left(t\right)$ for the solution to the TE. Unfortunately, we cannot find an analytical expression for the Wiener process, so we perform its calculations numerically. Nevertheless, in Appendix C, we present a cumulant perturbation approach for τ≪1.

In Figure 1, we show $\Delta S\left(t\right)=S\left(t\right)-S\left(0\right)$ for the TE process and compare this plot against the analytical result for the Wiener case: $S_{W}\left(t\right)=\ln\left(\sqrt{4\pi eDt}\right).$ In all cases, we use a thin Gaussian as the initial condition: ${\left.\psi \left(x,t\right)\right|}_{t=0}=\exp\left(-x^{2}/2\sigma^{2}\right)/\sqrt{2\pi \sigma^{2}}$ with dispersion σ=0.07. The parameters of the TE are τ=5 and v=1, and for the Wiener process, we take D=5.

In Figure 2, we show $I\left(t\right)$ versus $S\left(t\right)$ for the solution to the TE. This plot also shows a linear fit (in red) at short times 0≤t⪅10, while at long times, the behavior is non-linear. The parameters of the TE are as in Figure 1, i.e., τ=5 and v=1.

In Figure 3, we show the complexity measure $C\left(t\right)$ for the solution to the TE. While this measure for the Wiener process is a constant as shown in (Equation 11), for the TE, this measure is a non-trivial function of time. Parameters of the TE are as in Figure 1 and Figure 2, i.e., τ=5 and v=1. This complexity $C\left(t\right)$ shows a maximum at time $t_{\max}\approx 25$. This time is one order of magnitude larger than the relaxation time $t_{\mathrm{Relax}}\approx 2\tau$ in the TE; see Appendix A. In Figure 4, we show the complexity measure $C\left(t\right)$ for the TE, with parameters τ=1 and v=1. This complexity $C\left(t\right)$ shows a narrow peak and a maximum at time $t_{\max}\approx 4$. The time, $t_{\max}$, at which the complexity attains its maximum value, is described by the following relation:(19)$$\frac{d}{dt}\ln\frac{1}{\sqrt{I_{\mathrm{TE}}\left(t\right)}}=\frac{d}{dt}S_{\mathrm{TE}}\left(t\right).$$

This implies that when the relative velocity of Fisher’s information, $\dot{I}\left(t\right)/I=2\left|\dot{S}\left(t\right)\right|$, equals twice the modulus of the Entropy velocity, an observation emerges. In hyperbolic diffusion, the loss of information and the increase in disorder are not equivalent measures. This issue is connected to the fact that the solution of the TE has two quite different behaviors in the short and long times, as seen in Section A.2.

In Figure 5a,b, we show the behavior of the complexity $C\left(t\right)$. That is, we plot the maximum value $C\left(t_{\max}\right)$ and the time $t_{\max}$ as a function of τ. These values can be fitted by the following expressions:(20)$$\begin{array}{ccc} t_{\max} & = & -0.77549+5.1738\;\tau \end{array}$$(21)$$\begin{array}{ccc} C\left(t_{\max}\right) & = & -0.18314+8.2431\;\tau +70.079\;\tau^{2} \end{array}$$

For τ≫1, we can use the toy model (Equation 32) to characterize the ballistic regime. Then, it is possible to calculate $t_{\max}$ as a function of all the physical parameters. Using (Equation 31) and (Equation 45), we obtain the following:(22)$$t_{\max}≃3\tau +\tau \log\left[-3\exp\left[-3+\sigma^{2}/2{\left(v\tau \right)}^{2}\right]\right]-\sigma^{2}/2{\left(v\tau \right)}^{2}$$

In Figure 6, we show the Cramér–Rao inequality (Equation 14) applied to the solution of the TE under the parameters τ=5 and v=1. The plot shows the time-dependent behavior of the product $I\left(t\right)K_{2}\left(t\right)$. It is to be noted that when the initial condition for the solution to $\psi \left(x,t\right)$ is not a delta function, as in (Equation 3), the second cumulant $K_{2}\left(t∼0\right)$ does not go to zero. In fact, the second cumulant is given by (Equation 50) and goes to a constant for t→0. In this case, Fisher’s information does not diverge at t=0. This issue can be seen in the logarithmic representation in Figure 6 (thus, $\lim_{t\to 0}I\left(t\right)K_{2}\left(t\right)\to 1$). In the same Figure, we plot the Cramér–Rao inequality, calculated analytically with our toy model based on the approximation (Equation 32), which is valid for τ≫1. The Cramér–Rao inequality tends, for t→∞, to the lower bound one, corresponding to the Wiener process, as shown in (Equation 13).

We note that the convergence in time from the solution to the TE toward the Wiener process can also be proved by calculating the time-dependent kurtosis, as shown in Section A.3. In addition, in the limit τ≪1, a perturbation theory can be presented in terms of all cumulants of $\psi \left(x,t\right)$. Therefore, for example, Shannon’s entropy for the TE can be calculated, as shown in Appendix C.

## 5. Conclusions

The Wiener process is ubiquitous in nature because it is a time-dependent Gaussian process. For the Wiener process, the complexity *C* and the Cramér-Rao bound are well known, while if the diffusion process has a finite velocity of propagation, the situation is less known. In the present work, we studied the time-dependent Fisher $I_{\mathrm{TE}}\left(t\right)$ and Shannon $S_{\mathrm{TE}}\left(t\right)$ functionals associated with the hyperbolic diffusion, as characterized by the telegrapher’s equation. The solution to the telegrapher’s equation shows ballistic behavior at a short time (t≪τ), while at a long time (t≫τ), the behavior is diffusive, and the crossover between both regimes is of the order of the relaxation time, τ. Therefore, it is important to know how information measures behave in a hyperbolic diffusion protocol. In this context, we characterized the complexity $C\left(t\right)$ of this hyperbolic diffusion process as a function of time, showing that this measure has a maximum at $t_{\max}$ before relaxing to a constant value. We also presented the response of this time $t_{\max}$ as a function of the parameter τ ($\tau^{-1}$ is the rate of dissipation). The same occurs with the Cramér–Rao bound connecting Fisher’s information and the spatial uncertainty $\Delta x{\left(t\right)}^{2}$ in the process. A relation between a *non-local x*-parameter Fisher’s information with the *local t*-parameter Fisher’s information has been established; see (Equation 40), as well as the time-behavior of the Fisher and Shannon functionals for the solution to the telegrapher’s equation. A toy model approximation, for large τ, was used to calculate analytically the Cramér–Rao inequality as a function of time, as well as the relative entropy in the ballistic regime, see Figure 1.

Numerical results have been used to study the Fisher and Shannon functionals as functions of the parameter τ and time *t*. In addition a perturbation approach (in cumulants series) for small τ for the solution to the hyperbolic diffusion is also presented in Appendix C. This perturbation is a useful tool to analytically approximate many statistical objects in the theory of information.

For many decades, Fisher’s information has been used to find limits on the precision of codes and protocols. On the other hand, the telegrapher’s equation has found applications in many areas of interest where finite-velocity diffusion is the crucial ingredient, such as in engineering code problems, the transport of electromagnetic signals, biophysics of neuronal responses, machine learning, etc. We believe that the present work will stimulate research in the area of the theory of information on hyperbolic diffusion.

Extensions of the present approach can also be conducted when the rate of absorption of energy (characterized by the parameter $\tau^{-1}$) has time fluctuations (noise) [11]. In this case, the general interest lies in averaging statistical objects over realizations of disorder (time fluctuations in the rate $\tau^{-1}$). Works in this direction are in progress.

Our results can also be extended to consider the issue of random initial conditions in the telegrapher’s equation and to model temperature fluctuations. Therefore, the present approach can help in estimating and statistically inferring physical parameters derived from cosmic microwave background data [17].

## Figures and Tables

**Figure 1 entropy-25-01627-f001:**
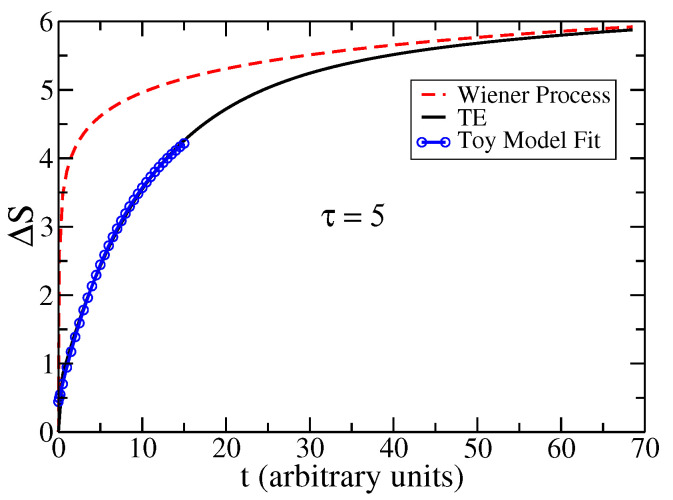
$\Delta S\left(t\right)$ as a function of time for a Wiener process (red dashed line), and the solution to the TE (full black line). The blue circle line represents $\Delta S\left(t\right)$ from the toy model solution to the TE, the fit is valid for t/τ≪1; see Section B.1. The parameters are τ=5, v=1.

**Figure 2 entropy-25-01627-f002:**
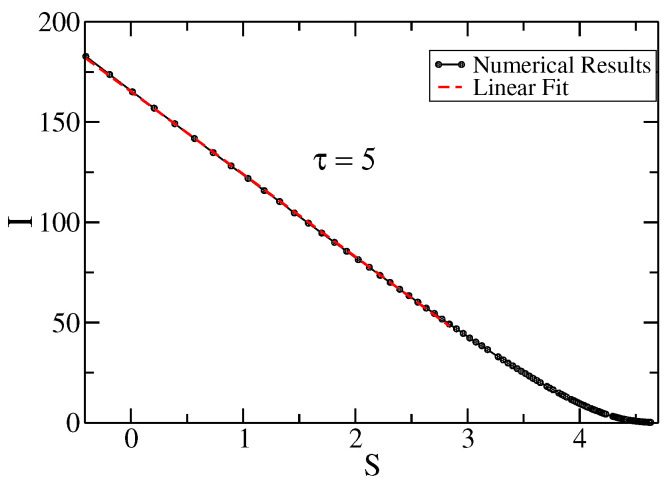
Fisher’s information as a function of entropy for the solution to the TE (black circle). The linear fit (for some values of t≲10) is denoted by the dashed red line. The parameters are τ=5, v=1. The equation of linear fit is I(S)=−41.57S+166.

**Figure 3 entropy-25-01627-f003:**
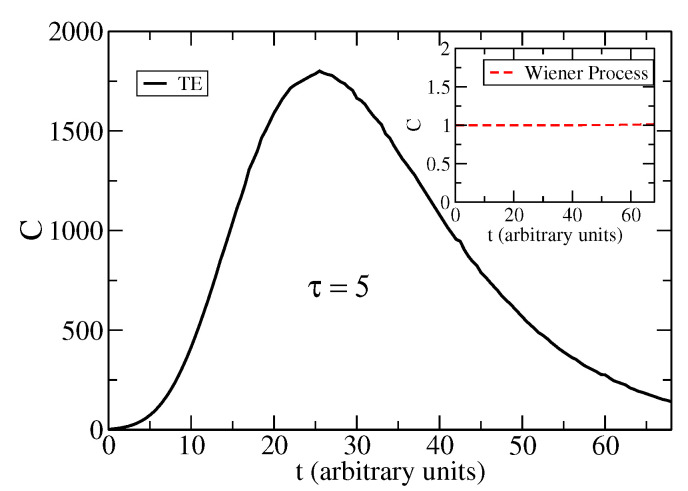
Complexity $C\left(t\right)$ as a function of time for the solution to the TE (full black line). The inset corresponds to the Wiener process (red dashed line). The parameters are τ=5, v=1.

**Figure 4 entropy-25-01627-f004:**
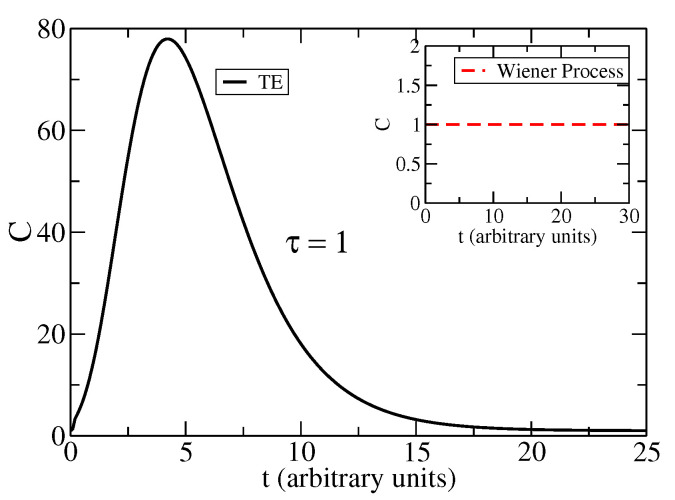
Complexity $C\left(t\right)$ as a function of time for the solution to the TE (full black line). The inset corresponds to the Wiener process (red dashed line). The parameters are τ=1, v=1.

**Figure 5 entropy-25-01627-f005:**
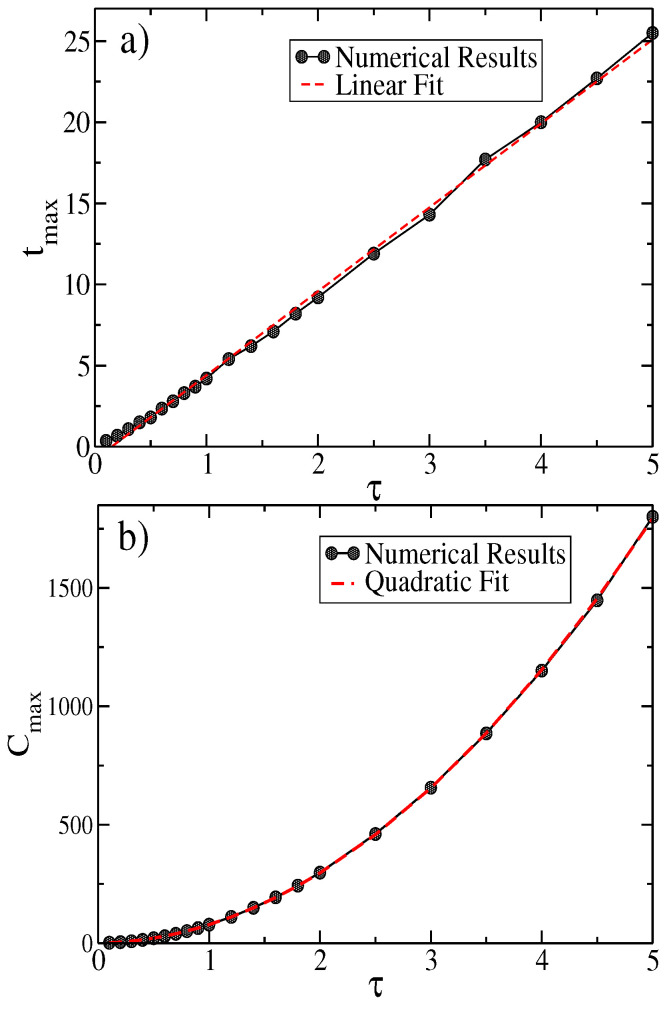
Characteristics of the complexity measure $C\left(t\right)$. (**a**) Maximum time $t_{\max}$ as a function of τ. (**b**) Maximum value for the complexity as a function of τ.

**Figure 6 entropy-25-01627-f006:**
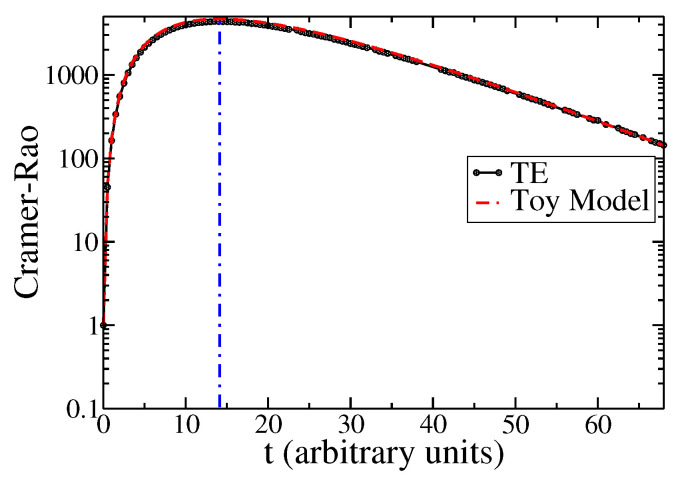
Cramér–Rao inequality as a function of time for the solution to the TE (full black line with circles), and the attenuated wave: $\psi \left(x,t\right)\approx e^{-x/L}\exp\left(-{\left(x-tv\right)}^{2}/2\sigma^{2}\right)/\sqrt{2\pi \sigma^{2}}$, (toy model (Equation 32) in red dashed line). The vertical blue dashed line represents the location of the $t_{\max}$ given by our formula (Equation 22). The parameters are τ=5, v=1, σ=0.07.

## Data Availability

The data that support the findings of this study are available from the corresponding author upon reasonable request.

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
