# Peer review of "Fisher and Shannon Functionals for Hyperbolic Diffusion"

_entropy, 2023, doi:10.3390/e25121627_

Round 1

Reviewer 1 Report

Comments and Suggestions for Authors

The Authors consider an interesting problem of telegrapher's equation and the present manuscript contains new results on the Fisher information, the Shannon entropy and the Cramer-Rao inequality. The material presented in the manuscript appears to be sound and of current interest to the scholars working in the field. I found the obtained results fresh and different from many papers related to finite-velocity diffusion problems and I hope that they will open new vistas in the field. Therefore, I recommend publication of the current manuscript in the journal Entropy.  

Author Response

Response to the First Referee

We thank very much the comments of the First Referee. 

Reviewer 2 Report

Comments and Suggestions for Authors

See my report

Comments on the Quality of English Language

See my report

Author Response

List of changes

    The first paragraph in the Section I has been enlarged to broadening the perspective of our paper. As suggested by Referee 3, the last paragraph in this Section I is new (in this paragraph we have clarified the organization and novelty of our work).
    As required by the Referee 3, the two paragraphs after Eq. (22), has been enlarged in order to clarify our Figure 6, and the necessity of our Appendices.
    As suggested by Referee 2, last paragraph in Section V is new. In this paragraph we have incorporated few words to remark how our approach can be useful on real applications in the context of the statistical inference in CMB data.
    References [17, 18, 32] are new.
    The Figure 1 has been modified in according with the suggestion of the Referee 3.
    English, all along the paper, has been checked.
    As suggested by Referee 3 the Toy Model has been better introduced in Appendix A.1

Response to the third Referee

“I am familiar with a few concepts treated in this paper but not all. In particular, I don’t know the telegrapher’s equation, hyperbolic diffusion, and the wave aspect, being better in discrete time series.”… “More generally, I am not able to distinguish what is new in this paper”…  “The form of the paper should also be improved. There are many mistakes of all kinds. See the detailed remarks below”…

We understand that due to the fact that the third Referee is not familiar with the TE, he/she has made many comments/questions, which we have considered in the new version of our manuscript. Nevertheless, some comments cannot be considered, so we have answered these questions bellow:

    1, Section I, Eq. (1). “v” is undefined. I suppose it is a velocity but it should be said

The parameter “v” was originally well defined in the line after Eq. (1).

    2, Section I, line +2. Is this a Wiener process in your context? I know what is a Wiener stochastic process but it is not the same. Is there a relation?

Of course there is a clear/unique relation between any realizations of the Wiener (stochastic) process and the PDF of the diffusion equation (Brownian motion).

    2, Section I, line +9. The existence of these appendices should be mentioned in the introduction and before the first reference. Why is it in the appendices? Is the material new? Probably not since there are references to [1] and [29]. More generally, I am not able to distinguish what is new in this paper. It should be highlighted in the introduction.

    I) We have enlarged the Section I to remark the novelty of our work, and the necessity to leave all the algebra to the appendices in order to simplify the lecture of the manuscript.
    II) We have relocated some references, in order to avoid confusions considering a non-specialized audience.

p.2, Section II, line +2. “concerned on” should read “concerned with”. Please add a reference. Why “therefore”?

The hyperbolic (diffusion) equation (1) can be used for wave-like descriptions (as pointed in the introduction), or for distribution problems (positive and normalized PDF). This is just the purpose of the present work, therefore it is necessary to introduce conditions pointed out in Eq. (2).

p.2, Section II, line +8. Is this new? Otherwise, please provide a reference.

Yes it is new, and it is now better explained in the introduction Section I.

p.2, Section II, line +11. This is not clear. What do you mean? What is a “bullet initial condition”?

It was explained in the original version. Now it is remarked after Eq. (4).

    2, Section III.A, line +1. Is the material in Section III.A new? On the contrary, please provide a reference. I would say no since you refer to [27, 28] for the definition of complexity.

Yes it is new, and it is now explained in the introduction section I. References have been relocated.

    2, Section III, line +6. I don’t understand the sentence with dimensionless and the next one too.
    3, Section III.A, line +4. What is the relation between “SW” and “S”?
    I) In physics research we use distributions (PDF) with given dimensions (i.e.: space, time, etc), so our necessity to consider dimensionless quantities in the argument of a logarithm.
    II) SW (Shannon’s entropy of the Wiener process), and S (generic Shannon’s entropy), SAW (entropy for the attenuated wave), etc.

    3, Section III.A, line −2. Why “universal manner”? Also, I didn’t understand the next sentence.

We clarify this paragraph.

    4, Section III.B, line −7. Omit “:” and the equality is not nice (the same function with 1 and 2 arguments).

Eq. (15) shows that Fisher’s information depends on both times (t,t’), this is due to the non-Markov kernel in Eq.(4).    

    5, Figure 1. The blue line in Figure 1 is difficult to see. It would be better to use a red color. Why mention “arbitrary units”? Why write “(Color on-line)” in each picture? I didn’t understand the toy model.
    I) In physics research it is usual to plot with arbitrary units of space, time, etc.
    II) We apology and we delete the sentence “color- on-line”.

III) Our “Toy model” is based in a short-time approximation for the solution of the TE (1); by the way, this was originally pointed out in Appendix A.1.

    5, Section IV, line +1. Avoid an abbreviation in the title. How were these results obtained?

By integrating numerically Eq.(1). This is now emphasized in the corresponding paragraph.

    6, Section IV, line −7. I could not understand several sentences like the one starting with “This means” two lines after Eq. (19). Why use a fit like in Figure 5?
    I) Formula (19) tells that the rate of decrease of Information and the rate of increase of Disorder are not equals.
    II) When it is not possible to have an analytical formula for C(t), it is nice and important to know the behaviour of C(t) in some regimes.

    7, Section IV, line −7. Why “Log-natural”?

To show (just in only one Figure) that for t --> 0 the Cramer-Rao bound goes to 1.

Reviewer 3 Report

Comments and Suggestions for Authors

This is a nice short paper about deriving information functionals for hyperbolic diffusions. The study of finite velocity processes driven by telegraph equation is a very well known field yet few studies considered these problems. So I like this paper. I have to say though that the authors looks at these functionals only from the perspective of information theory while there is also a line of work conducted by other authors looking at statistical inference for discretely observed such processes that the authors of the present work do no seem to be aware of. I suggest the authors to consider also this perspective and browse the literature on inference for such processes and add a paragraph or two of how these results can be useful in the context of statistical inference.

Another paragraph on real applications (not just simulated toy models) would probably broaden the scope and interest of this paper.

I have selected major revision below, just because there is no way to suggest adding content in the minor revision option. But my point is that little work is needed to improve this work by adding reference to statistical inference and real potential applications.

Author Response

Response to the Second Referee

"I have selected major revision below, just because there is no way to suggest adding content in the minor revision option. But my point is that little work is needed to improve this work by adding reference to statistical inference and real potential application"...

We thank the comments made by the 2nd Referee,

We have enlarged the Introduction Section I in order to point out another very interesting application of the TE (which in fact we did not the Spherically Hyperbolic Diffusion with applications to the CMB.

We have also enlarged the Conclusion Section V in order to remark that our approach can also be used to tackle the problem of random initial condition and therefore the inference analysis with real potential applications in astrophysics. Concerning to these comments, we have incorporated new references 17 and 18.

Round 2

Reviewer 2 Report

Comments and Suggestions for Authors

See my report

Comments on the Quality of English Language

See my report.

Author Response

List of changes

  • The Abstract and the core of the paper have been modified in accordance with the English revision made by the last referee and by a native English writer.
  • As required by the last referee we have reduced self-citations (now the proportion is 21%). We believe that the rest are important references to mention in the Introduction Section to quote the telegrapher’s equation in very different research areas. This issue increases the interest of the present paper for a larger audience in the Entropy Journal.
  • We have controlled the bibliography presentation according to the Journal instructions.
  • Old reference [32] has been divided, Ref. 4 is new, and Ref. 38 has been moved from the core of the text to the bibliography list, as requested by the referee.

Response to the referee

“Thank you for the new version and the answers. The link between differential equations and the Wiener process is not very clear to me but thank you for considering my remarks.”…

1) The link between a PDE and the Wiener process can be found in many book references (see a pedagogical presentation, using a functional approach, on pp.111, Ref. 30).

“The form of the paper should also be improved. There are many mistakes of all kinds. See the detailed remarks below. Most mistakes were detected by using a free grammar checker, but I would recommend making the paper read by a native English writer”…

2) We thank the Referee for his/her job in marking our English mistakes. We have considered all comments.

“p. 1, Section I, line +3. What is the hyperbolic equation? Why that name?”…

3) In our manuscript we wrote: hyperbolic diffusion equation, this name is associated with a process with finite-velocity diffusion. There are many references to this name: ranging from mathematics (Refs. [4, 33], etc.) to physics (see Refs [17, 18], etc.).

Reviewer 3 Report

Comments and Suggestions for Authors

thanks for taking into considerations adding references. The statistical inference part is still missing, but it is probably the scope of another paper and the authors do not seem to be familiar with the vast statistical literature of inference for stochastic processes and that particular on estimating/fitting these models from real discretely observed data.

Author Response

List of changes

  • We have incorporated minor English revisions following the feedback from Referee 2.
  • In response to the feedback from Referee 2, we have further reduced the self-citations to 17%. The remainingpapers are important references that are mentioned in the Introduction Section support the topic extensively. Unfortunately, we have had to delete three former self-citations: Ref [13] “on neutron diffusion”, Ref. [22] “on gravity waves on a random bottom”, Ref. [10] “on the localization of waves with random absorption”, following the referee’s suggestions.
  • Following Referee 2, the last paragraph in the Introduction Section emphasizes the novelty of the presented material.
  • Following Referee 3, the Section III.A is new. Here we have added a few words concerning the inference theory.
  • References [32, 33] are new.

           Response to the third referee

“Thanks for taking into considerations adding references. The statistical inference part is still missing, but it is probably the scope of another paper and the authors do not seem to be familiar with the vast statistical literature of inference for stochastic processes and that particular on estimating/fitting these models from real discretely observed data”…

You are correct that the analysis of statistical inference is beyond the scope of the current paper. Additionally, we are not very familiar with the literature on this subject. However, in response to the referees' remarks, we have added a new section III.C to briefly introduce the topic and include some new references.

Round 3

Reviewer 2 Report

Comments and Suggestions for Authors

See my report

Comments on the Quality of English Language

See my report

Author Response

List of changes

  • We have incorporated minor English revisions following the feedback from Referee 2.
  • In response to the feedback from Referee 2, we have further reduced the self-citations to 17%. The remainingpapers are important references that are mentioned in the Introduction Section support the topic extensively. Unfortunately, we have had to delete three former self-citations: Ref [13] “on neutron diffusion”, Ref. [22] “on gravity waves on a random bottom”, Ref. [10] “on the localization of waves with random absorption”, following the referee’s suggestions.
  • Following Referee 2, the last paragraph in the Introduction Section emphasizes the novelty of the presented material.
  • Following Referee 3, the Section III.A is new. Here we have added a few words concerning the inference theory.
  • References [32, 33] are new.

 Response to the second referee

 “If the percentage of self-citations was very slightly reduced, it is mainly because of a few justified additions. It seems that only the former [13] was omitted. Since a large part of the references in the first two paragraphs are already cited by the paper with the same first two authors [12], I suggested referring to the references in [12]. The authors did not follow that suggestion. On the other side, the lack of homogeneity of the bibliography seems now corrected”….

In our previous revision of the manuscript we deleted references [13, 22]. Now, as demanded by the referee, we have also deleted former reference [10]. However, we are not completely in agreement with the comment. Please see our list of changes.

“I asked about the novelty of the material. You answered positively in the reply of Revision 1 but there is no indication of this assertion of the paper. As you imagine, I am not able to detect that the results have appeared elsewhere before. Of course, I had no time to try to understand the contents of the paper”…

We have expanded the last paragraph of Section I in order to clearly state the novelty of all our results in the manuscript. Even though we consider these comments unnecessary for a specialized audience on the topic of TE.

“There are still a few mistakes but much less. See the detailed remarks below”…

We thank the referee for his/her work in pointing out our English mistakes. We have taken all of the comments into consideration.
